# Understanding and Improving Encoder Layer Fusion in Sequence-to-Sequence Learning

**Xuebo Liu**[1][*]**, Longyue Wang**[2]**, Derek F. Wong**[1]**, Liang Ding**[3]**, Lidia S. Chao**[1] **& Zhaopeng Tu**[2]
[1]NLP[2]CT Lab, Department of Computer and Information Science, University of Macau
[2]Tencent AI Lab [3]The University of Sydney
`nlp2ct.xuebo@gmail.com, {vinnylywang,zptu}@tencent.com,`
`{derekfw,lidiasc}@um.edu.com, ldin3097@sydney.edu.au`

## ABSTRACT

Encoder layer fusion (EncoderFusion) is a technique to fuse all the encoder layers (instead of the uppermost layer) for sequence-to-sequence (Seq2Seq) models, which has proven effective on various NLP tasks. However, it is still not entirely clear why and when EncoderFusion should work. In this paper, our main contribution is to take a step further in understanding EncoderFusion. Many of previous studies believe that the success of EncoderFusion comes from exploiting surface and syntactic information embedded in lower encoder layers. Unlike them, we find that the encoder embedding layer is more important than other intermediate encoder layers. In addition, the uppermost decoder layer consistently pays more attention to the encoder embedding layer across NLP tasks. Based on this observation, we propose a simple fusion method, SurfaceFusion, by fusing only the encoder embedding layer for the softmax layer. Experimental results show that SurfaceFusion outperforms EncoderFusion on several NLP benchmarks, including machine translation, text summarization, and grammatical error correction. It obtains the state-of-the-art performance on WMT16 Romanian-English and WMT14 English-French translation tasks. Extensive analyses reveal that SurfaceFusion learns more expressive bilingual word embeddings by building a closer relationship between relevant source and target embeddings. Source code is freely available at `https://github.com/SunbowLiu/SurfaceFusion`.

## 1 INTRODUCTION

Sequence-to-Sequence (Seq2Seq) learning (Sutskever et al., 2014) has advanced the state of the art in various natural language processing (NLP) tasks, such as machine translation (Bahdanau et al., 2015; Vaswani et al., 2017; Wu et al., 2019), text summarization (Wang et al., 2019b; Zhang et al., 2020), and grammatical error correction (Kiyono et al., 2019; Kaneko et al., 2020). Seq2Seq models are generally implemented with an encoder-decoder framework, in which a multi-layer encoder summarizes a source sequence into a sequence of representation and another multi-layer decoder produces the target sequence conditioned on the encoded representation.

Recent studies reveal that fusing the intermediate encoder layers (EncoderFusion) is beneficial for Seq2Seq models, such as layer attention (Bapna et al., 2018), layer aggregation (Dou et al., 2018; Wang et al., 2019c), and layer-wise coordination (He et al., 2018). Despite its effectiveness, not much is known about how fusing encoder layer representations work. The intuitive explanation is that fusing encoder layers exploits surface and syntactic information embedded in the lower encoder layers (Belinkov et al., 2017; Peters et al., 2018). However, other studies show that attending to lower encoder layers (excluding the encoder embedding layer) does not improve model performance (Domhan, 2018), which is conflicted with existing conclusions. It is still unclear why and when fusing encoder layers should work in Seq2Seq models.

This paper tries to shed light upon behavior of Seq2Seq models augmented with EncoderFusion method. To this end, we propose a novel *fine-grained layer attention* to evaluate the contribution of

---

[*]Work was done when Xuebo Liu and Liang Ding were interning at Tencent AI Lab.

individual encoder layers. We conduct experiments on several representative Seq2Seq NLP tasks, including machine translation, text summarization, and grammatical error correction. Through a series of analyses, we find that the uppermost decoder layer pays more attention to the encoder embedding layer. Masking the encoder embedding layer significantly drops model performance by generating hallucinatory (i.e. fluent but unfaithful to the source) predictions. The encoded representation of the standard Seq2Seq models (i.e. w/o fusing encoder layers) may not have enough capacity to model both semantic and surface features (especially at the encoder embedding layer). We call the problem described above the *source representation bottleneck*.

Based on this observation, we simplify the EncoderFusion approaches by only connecting the encoder embedding layer to softmax layer (*SurfaceFusion*). The SurfaceFusion approach shortens the path distance between source and target embeddings, which can help to learn better bilingual embeddings with direct interactions. Experimental results on several Seq2Seq NLP tasks show that our method consistently outperforms both the vanilla Seq2Seq model and the layer attention model. Extensive analyses reveal that our approach produces more aligned bilingual word embeddings by shortening the path distance between them, which confirm our claim.

Our main contributions are as follows:

- We introduce a *fine-grained layer attention* method to qualitatively and quantitatively evaluate the contribution of individual encoder layers.
- We demonstrate that the encoder embedding layer is essential for fusing encoder layers, which consolidates conflicted findings reported by previous studies.
- We propose a simple yet effective *SurfaceFusion* approach to directly exploit the encoder embedding layer for the decoder, which produces more expressive bilingual embeddings.

## 2 PRELIMINARIES

### 2.1 SEQUENCE-TO-SEQUENCE LEARNING

Seq2Seq learning aims to maximize the log-likelihood of a target sequence $\mathbf{y} = \{y_1, \ldots, y_J\}$ conditioned on a source sequence $\mathbf{x} = \{x_1, \ldots, x_I\}$, which is formulated as: $\hat{\mathbf{y}} = \arg\max \log P(\mathbf{y}|\mathbf{x})$. Typically, Seq2Seq learning can be implemented as various architectures (Bahdanau et al., 2015; Gehring et al., 2017; Vaswani et al., 2017; Wu et al., 2019), among which the Transformer (Vaswani et al., 2017) has advanced the state of the art. Without loss of generality, we introduce Transformer as the testbed in this paper. Transformer consists of an encoder $\mathcal{E}$ equipped with $N$ identical layers to map the source sequence $\mathbf{x}$ into distributed representations, based on which a decoder $\mathcal{D}$ equipped with $M$ identical layers generates the target sequence $\mathbf{y}$:

$$\mathbf{X}^N = \mathcal{E}(\mathbf{X}^0) \quad \overset{N}{\underset{n=1}{:=}} \quad \text{FFN}\left(\text{ATT}(\mathbf{X}^{n-1}, \mathbf{X}^{n-1}, \mathbf{X}^{n-1})\right) \tag{1}$$

$$\mathbf{Y}^M = \mathcal{D}(\mathbf{Y}^0, \mathbf{X}^N) \quad \overset{M}{\underset{m=1}{:=}} \quad \text{FFN}\left(\text{ATT}\left(\text{ATT}(\mathbf{Y}^{m-1}, \mathbf{Y}^{m-1}, \mathbf{Y}^{m-1}), \mathbf{X}^N, \mathbf{X}^N\right)\right) \tag{2}$$

where $\mathbf{X}^0$ denotes the sum of the word embeddings $\mathbf{X}_{\text{emb}}$ and position embeddings $\mathbf{X}_{\text{pos}}$ of $\mathbf{x}$, $\mathbf{Y}^0$ denotes that of the shifted right $\mathbf{y}$, $\text{FFN}(\cdot)$ denotes a position-wise feed-forward network, and $\text{ATT}(\cdot)$ denotes a multi-head dot-product attention network with three arguments–query, key and value. Residual connection (He et al., 2016) and layer normalization (Ba et al., 2016) are used in each sub-layer, which are suppressed in Equation 1 and 2 for clarity. Finally, the output representation $\mathbf{Y}^M$ of the decoder is projected into the probability $P(\mathbf{y}|\mathbf{x})$, which is optimized during model training.

### 2.2 EXPERIMENTAL SETUP

To validate the universality of source representation bottleneck in Seq2Seq models, we conducted experiments on three representative tasks, which vary from the distance between input and output domains and the scale of training data:

**Machine translation** takes a sentence in one language as input, and outputs a semantically-equivalent sentence in another language. We conducted experiments on three benchmarking datasets: small-scale WMT16 Romanian-English (Ro-En; 0.6M instances), medium-scale WMT14 English-German

(En-De; 4.5M instances), and large-scale WMT14 English-French (En-Fr; 36.0M instances). The tokenized BLEU score (Papineni et al., 2002) was used for all the translation tasks.

**Text summarization** takes a long-text document as input, and outputs a short and adequate summary in the same language. We used the CNN/Daily Mail corpus (0.3M instances). We evaluated with the standard ROUGE metric (Lin, 2004), i.e. Rouge-1, Rouge-2, and Rouge-L.

**Grammatical error correction** takes a sentence with grammatical errors as input, and outputs a corrected sentence. We used CONLL14 datasets as the testbed (1.4M instances). The MaxMatch ($M^2$) scores (Dahlmeier & Ng, 2012) were used for evaluation with precision, recall, and $F_{0.5}$ values.

The machine translation task has distant input/output domains (i.e. in different languages), while the other tasks have similar input/output domains (i.e. in the same language). We used Transformer (Vaswani et al., 2017) as the Seq2Seq model. Details of the datasets and model training are listed in Appendix A.1.

## 3    BEHAVIOR OF ENCODERFUSION

In this section, we first formulate our research hypothesis of *source representation bottleneck* (§3.1) that EncoderFusion expects to solve. In the following subsections, we propose a *fine-grained layer attention* model (§3.2) to validate our hypothesis on well-designed experiments (§3.3).

### 3.1    SOURCE REPRESENTATION BOTTLENECK

Seq2Seq models learn more abstract features with the increase of layer level (i.e. $\mathbf{X}^0 \to \mathbf{X}^N$ and $\mathbf{Y}^0 \to \mathbf{Y}^M$) (Belinkov et al., 2017). It has been extensively validated that a reasonable use of both the abstract representations (at higher-level layers) and the surface representations (at lower-level layers) is beneficial for various NLP (Lu & Li, 2013; Hu et al., 2014; Dou et al., 2018; Peters et al., 2018) and CV (Long et al., 2014; Pinheiro et al., 2016; Lin et al., 2017; Chen et al., 2018a) tasks.

However, the Seq2Seq decoder only takes the abstract representations at uppermost layer $\mathbf{X}^N$ as input (Equation 2), while ignores other usefully surface representations at other layers $\mathbf{X}^n$ ($n < N$). Although $\mathbf{X}^N$ has encoded surface features from low-level representations through layer-by-layer abstraction and residual connections, we hypothesize that its limited representation capacity may not sufficiently model those surface features from lower encoder layers, especially the embedding layer. We call such an issue as *source representation bottleneck*.

### 3.2    FINE-GRAINED LAYER ATTENTION

For each decoder layer, layer attention (Bapna et al., 2018; Peters et al., 2018) assigns normalized scalar weights to all encoder layers, providing a direct way for evaluating the contributions made by each encoder layer. However, the capacity of a simple scalar weight is limited, leading to insufficient evaluation of the contributions.

Motivated by fine-grained attention (Choi et al., 2018) that each element of a context vector receives an individual attention weight, we propose a *fine-grained layer attention* model to combine the advantages of both techniques. This allows us to more convincingly evaluate the contribution of individual encoder layer to the model performance. Besides, the nature of fine-grained attention enables us to give in-depth analyses of the representation power in §3.3.

Specifically, we replace the layer-agnostic source representation $\mathbf{X}^N$ with the layer-aware representation $\mathbf{S}^m$ for each decoder layer $\mathbf{Y}^m$, which is calculated as:

$$\mathbf{S}^m = \sum_{n=0}^{N} \hat{\mathbf{w}}^{m,n} \odot \mathbf{X}^n, \quad \hat{\mathbf{w}}^{m,n} = \left[ \hat{w}^{m,n,1}, \dots, \hat{w}^{m,n,D} \right], \quad \hat{w}^{m,n,d} = \frac{\exp(w^{m,n,d})}{\sum_{n'=0}^{N} \exp(w^{m,n',d})}$$

where $\odot$ denotes an element-wise multiplication, and $w^{m,n,d}$ denotes an element in the learnable attention weight $\mathbf{W} \in \mathbb{R}^{M \times (N+1) \times D}$, where $D$ is the dimensionality of the source representation. When $n = 0$, we use the word embeddings $\mathbf{X}_{\text{emb}}$ without position embeddings as $\mathbf{X}^0$, which has been empirically proved effective. We applied a regularization technique – DropConnect (Wan et al.,

2013) to the attention weight $\mathbf{W}$ for a stable training, which randomly drops each $w^{m,n,d}$ with a probability $p$ and divides $\mathbf{W}$ by $1 - p$. We set it to 0.3 for all the experiments.

Table 2 lists the results. The proposed fine-grained layer attention model consistently outperforms the vanilla Transformer across Seq2Seq tasks, demonstrating the benefit of fusing surface features at lower-level layers.

We evaluated several EncoderFusion methods in Table 1, including layer aggregation (Dou et al., 2018), layer-wise coordination (He et al., 2018), and coarse-grained layer attention (Bapna et al., 2018). Their results are respectively 34.05, 34.19, and 34.32, which are all lower than that of fine-grained layer attention (34.45). Based on these experimental results, we thus choose fine-grained layer attention as a representative of EncoderFusion in the following analyses.

Table 1: Results of existing encoder layer fusion methods on the WMT16 Ro-En translation task.

| Model | BLEU |
|---|---|
| Vanilla Transformer | 33.80 |
| Layer aggregation | 34.05 |
| Layer-wise coordination | 34.19 |
| Coarse-grained layer attention | 34.32 |
| Fine-grained layer attention | 34.45 |

### 3.3 BEHAVIOR CHANGES ACROSS ENCODER LAYERS

In this section, we investigate whether the surface features at lower encoder layers (especially the encoder embedding layer) contribute to the model performance via carefully designed experiments.

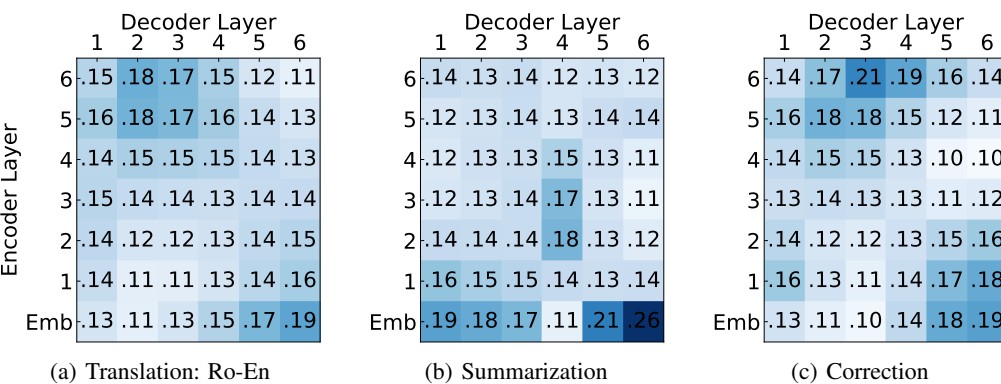

(a) Translation: Ro-En      (b) Summarization      (c) Correction

Figure 1: Attention distribution that each decoder layer ($x$-axis) attending to encoder layers ($y$-axis).

**Visualization of layer attention** We first visualize the learned layer attention distribution in Figure 1, in which each weight is the averaged attention weights over all dimensions. Generally, a higher weight denotes more contribution of an encoder layer to the corresponding decoder layer.

Clearly, in all tasks higher decoder layers especially the uppermost ones pay more attention to the encoder embedding layer, which indicates that the surface representations potentially bring some additional useful features to the model performance. Voita et al. (2019); Wang & Tu (2020) reveal that the upper layers of decoder are responsible for the translation part while the lower layers for the language modeling part. Similarly, our results show that surface representations might play an important role in learning to translate source tokens.

Among the Seq2Seq models, there are still considerable differences in the attention heatmaps. In the summarization model, almost all decoder layers focus more on the encoder embedding layer, while in the other two models the intermediate decoder layers pay more attention to the higher-level encoder layers. This is consistent with the findings of Rothe et al. (2019), in which they reveal that the summarization task, as a typical extractive generation task, tends to use more surface features to generate extractive summaries. In contrast, both machine translation and error correction tasks require a large amount of syntactic and semantic information, which are generally embedded in higher-level encoder layers (Peters et al., 2018).

However, we still cannot conclude that source representation bottleneck does exist in Seq2Seq models, since the surface features might act as a noise regularizer to improve the robustness of encoder output representations. To dispel the doubt, we further design two experiments to directly evaluate the effectiveness of surface features at the encoder embedding layer.

**Contribution of individual encoder layer**    In this experiment, we quantitatively analyze the behaviors change of a trained Seq2Seq model when masking a specific encoder layer (i.e. turning its attention weight to zero and redistribute the other attention weights). Note that the masking operation does not affect the information flow of encoding calculation, i.e. keeping Equation 1 unchanged.

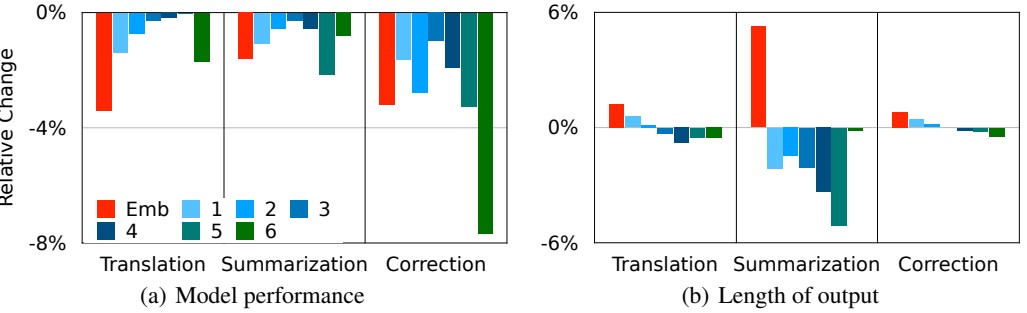

Figure 2: Relative changes of (a) model performance and (b) length of output when masking individual encoder layer in the trained Seq2Seq models. As seen, masking the embedding layer leads to a significant drop of model performance and increase of output length.

Figure 2(a) shows the contribution of individual encoder layer to model performance. As seen, masking the encoder embedding layer seriously harms the model performance in all tasks, which confirms our claim that the surface features in the embedding layer are essential to Seq2Seq models.

Figure 2(b) shows the results on the output length. Masking the encoder embedding layer consistently increases the length of generated output, which is especially significant for the summarization model. One possible reason is that the instances in translation and correction tasks have similar input/output lengths, while the summarization instances have distant input/output lengths.

By analyzing the model outputs, we found that the Seq2Seq models tend to generate some hallucinatory (i.e. fluent but unfaithful to the source) predictions (Lee et al., 2019; Wang & Sennrich, 2020) when masking the embedding layer. Taking the correction task for an example, a right prediction "anyone" was replaced by the hallucinatory prediction "friends of anyone" in the masked model, in which the corresponding source contains no information related to "friends". This issue becomes worse in the summarization task, since the hallucinatory prediction is more likely to be a sentence.

The additional hallucinations will increase the output length and reduce the model performance. In addition, Lee et al. (2019) point out that even if hallucinations occur only occasionally, the Seq2Seq model may evidently lose user trust than other prediction problems, indicating the importance to fuse surface features at the embedding layer. More cases are studied in Appendix A.2.

**Expressivity of attended dimensions in the encoder embedding layer**    As shown in Figure 1, the uppermost decoder layer pays most attention to the encoder embedding layer (i.e. the lower right corner). If the embedding layer acts as a noise regularizer, the layer dimensions would be randomly attended by the fine-grained model; otherwise, the dimensions of higher attention weights should be distinguished from the other dimensions.

Starting from this intuition, we reordered the dimensions of the encoder embedding layer according to the attention weights $\hat{\mathbf{w}}^{M,0}$, and split it into two equal sub-embedding matrices, i.e. *more attended dimensions* and *less attended dimensions*. We compared the expressivity of the two sub-embedding matrices by the commonly-used singular value decomposition (Gao et al., 2019; Wang et al., 2019a; Shen et al., 2020), in which higher normalized singular values denote that the embedding is more uniformly distributed, thus are more expressive. The singular values are normalized by dividing them by the largest value and their log scale values are reported for better clarity.

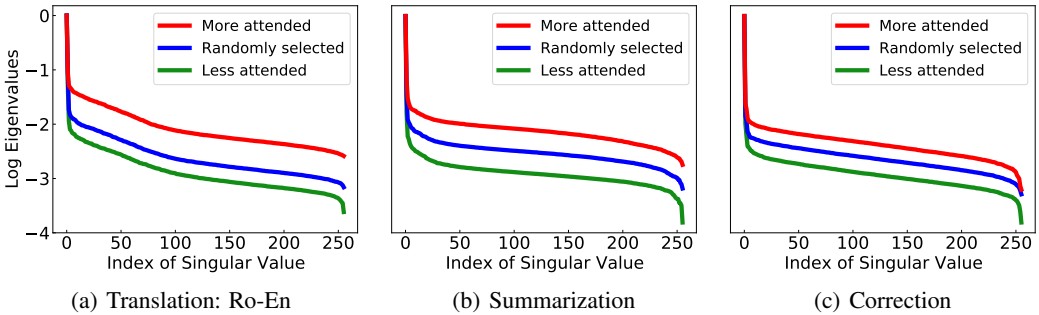

Figure 3: Log scale singular values of the three sub-embedding matrices in the fine-grained layer attention models. Higher log eigenvalues denote more expressivity of the dimensions.

Figure 3 depicts the singular value results. For comparison, we also report the values of the randomly selected dimensions. Clearly, the more attended dimensions are most expressive, while the less attended dimensions are least expressive. These results demonstrate that the fine-grained attention model indeed extracts useful surface information from the encoder embedding layer, which does not play the role of a noise regularizer.

From the above experiments, we prove that the encoder embedding layer indeed provides useful surface information, which is not fully exploited by the standard Seq2Seq models.

## 4 OUR METHOD

In Section 3, we show that the uppermost decoder layer requires more surface features for better representation learning. One possible reason is that the uppermost decoder layer is used for predicting individual target token, which naturally benefits from more token-level surface features than sequence-level abstract features. To validate this assumption, we simplify fine-grained layer attention that only the uppermost decoder layer can attend to the embedding layer and output layer of the encoder. Empirical results show that the simplified variant works on par with the original one, revealing that the surface features embed at the source embedding layer is expressive.

Although layer attention model partially alleviates source representation bottleneck, it potentially introduces unnecessary intermediate encoder representations. To address this gap, we propose to directly connect the decoder softmax layer and the encoder embedding layer with a simple *SurfaceFusion* method.

### 4.1 SURFACEFUSION

Seq2Seq learning aims to maximize the log-likelihood of a target sequence $\mathbf{y}$ given a source sequence $\mathbf{x}$. In practice, it factorizes the likelihood of the target sequence into individually token likelihoods:

$$\hat{\mathbf{y}} = \arg\max \prod_{j=1}^{J} \log P(y_j) = \arg\max \prod_{j=1}^{J} \log P(y_j | y_{<j}, \mathbf{x}) \tag{3}$$

We rewrite $P(y_j)$ as a fused probability with the second condition term $\mathbf{x}$:

$$\log P(y_j) = \Phi\big(P(y_j | y_{<j}, \mathbf{x}),\ P(y_j | \mathbf{x})\big) \tag{4}$$

where $\Phi(\cdot)$ is a fusion method that will be described later, and $P(y_j | \mathbf{x})$ is a probability conditioned on the source surface features. Specifically, we employ a multi-head dot-product attention network (Vaswani et al., 2017) with a decoder output representation $\mathbf{y}_j^M$ as a *query*, encoder output representations $\mathbf{X}^N$ as *keys* , and encoder surface representations $\mathbf{X}_{\text{emb}}$ as *values*, to calculate a surface representation $\mathbf{r}(y_j, \mathbf{x})$.

Then we use the pre-softmax weight $\mathbf{V} \in \mathbb{R}^{d \times |\mathcal{V}_y|}$ of the vanilla model to transform the surface representation $\mathbf{r}(y_j, \mathbf{x}) \in \mathbb{R}^d$ into a pre-softmax logit $\widetilde{\mathbf{r}}(y_j, \mathbf{x}) \in \mathbb{R}^{|\mathcal{V}_y|}$. The final surface constraint

probability is calculated as:

$$P(y_j|\mathbf{x}) = \frac{\exp(\mathbb{1}_{y_j}(\widetilde{\mathbf{r}}(y_j, \mathbf{x}))/\tau)}{\sum_{w \in \mathcal{V}_y} \exp(\mathbb{1}_w \widetilde{\mathbf{r}}(y_j, \mathbf{x})/\tau)} \tag{5}$$

where $\mathbb{1}_w(\cdot)$ denotes an index function to take the logit of a target token $y$, and $\tau$ denotes a softmax temperature parameter to control the smoothness of the probability distribution $P(y_j|\mathbf{x})$. As $\tau$ approaches to 0, the distribution tends to be an one-hot distribution representing the token of the maximum probability. The distribution becomes uniform at a higher $\tau$.

**Choices of fusion function $\Phi$** There are many variants of fusion methods (Gulcehre et al., 2015; Sriram et al., 2017; Stahlberg et al., 2018). The aim of this paper is not to explore this whole space but simply to show that two fairly straightforward implementations works well and that SurfaceFusion helps for sequence-to-sequence models:

*Hard fusion:* $\quad \Phi_{\text{hard}} = \lambda \log P(y_j|y_{<j}, \mathbf{x}) + (1 - \lambda) \log P(y_j|\mathbf{x})$ (6)

*Soft fusion:* $\quad \Phi_{\text{soft}} = \log(\text{softmax}(E(y_j|y_{<j}, \mathbf{x}) + \log P(y_j|\mathbf{x}))$ (7)

where $\lambda$ is a pre-defined interpolation weight, and $E(y_j|y_{<j}, \mathbf{x})$ is the pre-softmax logit of the probability $P(y_j|y_{<j}, \mathbf{x})$. Compared to hard fusion, soft fusion removes the need for manually setting the hyperparameter $\lambda$.

The proposed SurfaceFusion method is easy to use. There are only two additional hyperparameters, i.e. $\lambda$ (Equation 6) and $\tau$ (Equation 5). We find that $\lambda$ is sensitive to the corpus scale but insensitive to the relationship of input/output domain, which was set to 0.9 for the En-De, En-Fr and correction tasks, and 0.8 for the Ro-En and summarization tasks. For $\tau$, it was set to 5 for soft fusion and 1 for hard fusion across different benchmarks. We kept other settings all the same with the vanilla models. In practice, we observed an additional 10% inference latency with the introduction of SurfaceFusion.

## 4.2 EXPERIMENTAL RESULTS

Table 2: Results of the proposed SurfaceFusion methods on the Seq2Seq tasks. "FGLA" denotes fine-grained layer attention. The existing results are Ghazvininejad et al. (2019) for Ro-En, Ott et al. (2018) for En-De and En-Fr, Ott et al. (2019) for summarization, and Chollampatt & Ng (2018) for correction. All reported scores are the higher the better.

| | Translation | | | Summarization | | | Correction | | |
|---|---|---|---|---|---|---|---|---|---|
| | **Ro-En** | **En-De** | **En-Fr** | **RG-1** | **RG-2** | **RG-L** | **Prec.** | **Recall** | **$F_{0.5}$** |
| **Existing** | 34.0 | 29.3 | 43.2 | 40.1 | 17.6 | 36.8 | 65.5 | 33.1 | 54.8 |
| **Vanilla** | 33.8 | 28.9 | 43.4 | 40.4 | 17.7 | 37.2 | 64.7 | 33.2 | 54.3 |
| **FGLA** | 34.5 | 29.1 | 43.5 | 40.8 | 18.0 | 37.5 | **67.7** | 31.9 | 55.3 |
| **Hard fusion** | **35.1** | **29.5** | **43.9** | 40.9 | 18.2 | 37.7 | 67.0 | 34.4 | 56.3 |
| **Soft fusion** | 34.0 | 29.0 | 43.6 | **41.0** | **18.3** | **37.9** | 66.8 | **35.0** | **56.6** |

**Model performance** Table 2 lists the results of the proposed approach on different tasks. In addition to the vanilla Seq2Seq model ("Vanilla"), we also report the results of existing studies on the same datasets ("Existing") for better comparison. Our re-implementation of the vanilla models matches the results reported in previous works, which we believe make the evaluation convincing.

Clearly, the proposed fusion approaches outperform the baselines (i.e. "Vanilla" and "FGLA") in all cases, while there are still considerable differences among model variations. Hard fusion performs better on the translation tasks, while soft fusion is superior on the summarization and correction tasks. Unlike hard fusion that performs at the probability level, soft fusion performs at the logit level to provide an earlier and direct way for fusing surface features, which might be a better solution for the tasks with a similar input/output domain.

**Closeness of word embeddings** SurfaceFusion shortens the path distance between source and target embeddings, which can help to learn better bilingual embeddings with direct interactions. Table 3 shows the cosine similarities between the tied source and target embeddings on the Ro-En translation task.

In the experiment, we first train an additional aligner (i.e. fast-align (Dyer et al., 2013)) on the training corpus and use the alignment links to construct a word dictionary. The results calculated over the dictionary show that the relationship between the source and target embedding becomes much closer (i.e. high cosine similarities). This can help each other to learn better representations, and has been validated to be beneficial for Seq2Seq models (Press & Wolf, 2017; Liu et al., 2019).

Table 3: Cosine similarities between aligned source and target word embeddings. "All" and "Non-Shared" denotes keeping or removing the aligned pair when the source and target words are the same, which are easier to be aligned.

|  | All | Non-Shared |
|---|---|---|
| **Vanilla** | 0.602 | 0.338 |
| **SurfaceFusion** | **0.650** | **0.417** |

**Expressivity of word embeddings** In this experiment, we quantitatively evaluate the expressivity of the word embeddings learned by different models using the singular value decomposition. The related experimental details and executions are similar to that of Figure 3.

Figures 4 shows the results of the tied source and target embeddings on the Ro-En translation task. The word embeddings of the vanilla model have fast decaying singular values, which limits the representational power of embeddings to a small sub-space. The SurfaceFusion model slows down the decaying and the singular values become more uniformly distributed, which demonstrates that the fused surface features remarkably enhance the representation learning of embeddings. This provides a better starting point for the model to effectively extract surface and abstract features, which leads to an improvement of model performance.

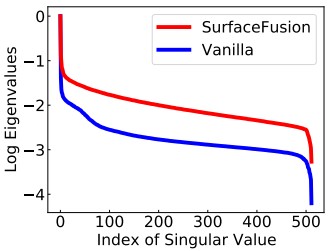

Figure 4: Log scale singular values of the embeddings.

## 5 RELATED WORK

**EncoderFusion in Seq2Seq** Lower encoder layers that embed useful surface features are far away from the training signals, which poses difficulty for deep Seq2Seq models to exploit such useful features. Although residual connections (He et al., 2016) have been incorporated to combine layers, these connections have been "shallow" themselves, and only fuse by simple, one-step operations (Yu et al., 2018). In response to this problem, several approaches have been proposed to fuse the encoder layers with advanced methods, such as layer attention (Bapna et al., 2018; Shen et al., 2018; Wang et al., 2019c), layer aggregation (Dou et al., 2018; Wang et al., 2018a; Dou et al., 2019; Li et al., 2020), and layer-wise coordination (He et al., 2018; Liu et al., 2020). Although these methods show promising results on different NLP tasks, not much is known about how the EncoderFusion works. In addition, some other studies show that exploiting low-layer encoder representations fail to improve model performance (Domhan, 2018).

In this paper, we consolidate the conflicting conclusions of existing studies by pointing out that the encoder embedding layer is the key, which can help Seq2Seq models to precisely predict target words. Based on this finding, we propose a novel *SurfaceFusion* to directly connecting the encoder embedding layer and the softmax layer, which consistently outperform current EncoderFusion approaches across different NLP tasks.

**Variants of Feature Fusion** Feature fusion aims to merge two sets of features into one, which is frequently employed in CV tasks, such as semantic segmentation (Long et al., 2014; Chen et al., 2018a; Zhang et al., 2018) and object detection (Pinheiro et al., 2016; Lin et al., 2017). Zhang et al. (2018) shows that simply fusing surface and abstract features tends to be less effective due to the gap in semantic levels.

For NLP tasks, researchers investigated fusion models for language understanding (Lu & Li, 2013; Hu et al., 2014; Peters et al., 2018) and language generation (Gulcehre et al., 2015; Sriram et al., 2017;

Stahlberg et al., 2018). Nguyen & Chiang (2019) propose to fuse features at representation-level, but we empirically find this kind of fusion method is not orthogonal to multi-layer models due to the large semantic gap. Gulcehre et al. (2015) combine the predictions produced by the Seq2Seq model and external LM predictions in a later fusion manner, which pose little impact to the original information flow. Stahlberg et al. (2018) improve upon it by removing the dependence on the manually defined hyper-parameter. In this work, we demonstrate the effectiveness of the two typical probability-level fusion methods on sequence-to-sequence learning tasks. Unlike them that rely on an external model, our approach only requires a surface attention module that can be jointly trained with the vanilla Seq2Seq model.

## 6 CONCLUSION AND FUTURE WORK

In this paper, we investigate how encoder layer fusion works on solving the source representation bottleneck. Based on a series of experiments on different Seq2Seq tasks, we find that the encoder embedding layer is important to the success of EncoderFusion by exploiting the useful surface information. Based on this observation, we propose a novel SurfaceFusion to directly connect the encoder embedding layer and softmax layer. Experiments show that SurfaceFusion consistently outperforms the conventional EncoderFusion in several datasets. Extensive analyses reveal that SurfaceFusion enhances the learning of expressive bilingual word embeddings for Seq2Seq models, which confirm our claim.

Future directions include validating our findings on more Seq2Seq tasks (e.g. dialogue and speech recognition) and model architectures (RNMT+ (Chen et al., 2018b) and DynamicConv (Wu et al., 2019)). It is also worthwhile to explore more alternatives to EncoderFusion from the perspective of exploiting the embedding layer.

## 7 ACKNOWLEDGMENTS

This work was supported in part by the Science and Technology Development Fund, Macau SAR (Grant No. 0101/2019/A2), and the Multi-year Research Grant from the University of Macau (Grant No. MYRG2020-00054-FST). We thank the anonymous reviewers for their insightful comments.

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

# A APPENDIX

## A.1 EXPERIMENTAL SETUP

Table 4: Statistics of the datasets and hyperparameters for the experiments. All the data have been tokenized and split into joint sub-word units (Sennrich et al., 2016). "Batch" denotes the number of source tokens and target tokens used in each training step. "DP" denotes the dropout value (Srivastava et al., 2014). "LP" denotes the length penalty (Wu et al., 2016). "Base" and "Big" denote the two kinds of model variants of Transformer. We chose the checkpoint with best validation ppl for testing.

| | Vocab | #Sents | | | Training | | | | Testing | |
|---|---|---|---|---|---|---|---|---|---|---|
| | Src/Tgt | Train | Dev | Test | Model | Batch | Step | DP | Beam | LP |
| **Ro-En** | 34,976 | 0.6M | 2K | 2K | Base | 16K | 60K | 0.3 | 4 | 1.0 |
| **En-De** | 32,768 | 4.5M | 3K | 3K | Big | 460K | 30K | 0.3 | 5 | 0.6 |
| **En-Fr** | 36,736 | 35.5M | 6K | 3K | Big | 460K | 80K | 0.1 | 5 | 0.9 |
| **CNN/DM** | 50,264 | 0.3M | 13K | 11K | Base | 64K | 70K | 0.1 | 4 | 2.0 |
| **CONLL** | 33,352 | 1.3M | 5K | 1K | Base | 64K | 80K | 0.2 | 6 | 0.6 |

**Machine translation** For WMT16 Romanian-English, we used the prepossessed data[1] and existing result from Ghazvininejad et al. (2019). The validation set is newsdev2016 and the test set is newtest2016. For WMT14 English-German, the prepossessed data[2] and existing result are derived from Ott et al. (2018). The validation set is newstest2013 and the test set is newstest2014. For WMT14 English-French, we reported the existing result from Ott et al. (2018) and followed them to preprocess the data sets. The validation set is newstest2012+2013 and the test set is newstest2014.

**Text summarization** For CNN/Daily Mail dataset, we used the existing result and preprocessing method of Ott et al. (2019). During testing, the minimum length was set to 55 and the maximum length was set to 140, which were tuned on the development data. We also followed Paulus et al. (2018) to disallow repeating the same trigram.

**Grammatical error correction** For CONLL14 benchmark, the preprocessing script[3] and existing result are given by Chollampatt & Ng (2018). We applied the regularization technique SwitchOut (Wang et al., 2018b) in this task to prevent overfitting, which was set to 0.8 for the source and 0.9 for the target.

Table 4 gives more details of the benchmarks. It is noted that other unmentioned hyperparameters keep the same with the original paper of Transformer (Vaswani et al., 2017). All the models are implemented by the open-source toolkit fairseq (Ott et al., 2019).[4]

## A.2 CASE STUDY

Tables 5, 6 and 7 give the cases from the three tasks. We can see that the hallucination issue related to surface features consistently appear over the different Seq2Seq tasks. The most representative cases are those from the correction task, in which very similar input/output sequences still make such mistakes.

Another observation is the *prediction omission* problem when masking the encoder output layer. The lack of abstract features leads to incomplete semantics of source representations, thus making Seq2Seq models omit generating a part of source, hurting the model performance. By looking at the cases over the three tasks, we find that the prediction omission is widespread in the prediction of modifiers, e.g. adjectives and adverbs.

---

[1] https://drive.google.com/uc?id=1YrAwCEuktG-iDVxtEW-FE72uFTLc5QMl
[2] https://drive.google.com/uc?id=0B_bZck-ksdkpM25jRUN2X2UxMm8
[3] https://github.com/nusnlp/mlconvgec2018/blob/master/data/prepare_data.sh
[4] https://github.com/pytorch/fairseq

Table 5: Examples from the Ro-En translation task. **Red** words are good predictions, while **blue** words are bad predictions. Masking the embedding layer ("Mask Emb") of the fine-grained layer attention model leads to hallucinatory predictions, prolonging the prediction length. While masking the output layer ("Mask Out") leads to prediction omissions, shortening the length.

| | | |
|---|---|---|
| *Hallucination* | Source | diseara voi merge acasa si voi dormi linistit . |
| | Reference | **i** will go home tonight and sleep well . |
| | Vanilla | **i** will go home and sleep quietly . |
| | Mask Emb | **the device** will go home and i will sleep peacefully . |
| | Mask Out | **i** will go home and sleep quietly . |
| *Omission* | Source | radem adesea mult atunci cand vorbim . |
| | Reference | we often **laugh a lot** when we talk . |
| | Vanilla | we often **laugh a lot** when we talk . |
| | Mask Emb | we often **laugh a lot** when we talk . |
| | Mask Out | we often **laugh** when we talk . |

Table 6: Examples from the CNN/DM summarization task.

| | | |
|---|---|---|
| *Hallucination* | Source | ... But it is able to carry just as much power - 400,000 volts . It is designed to be less obtrusive and will be used for clean energy purposes ... |
| | Reference | ... But it is able to carry just as much power - 400,000 volts . **It is designed to be less obtrusive and will be used for clean energy .** |
| | Vanilla | ... But it is able to carry just as much power - 400,000 volts . **It is designed to be less obtrusive and will be used for clean energy .** |
| | Mask Emb | ... It is able to carry just as much power - 400,000 volts . **The design is a T-shape , with two ' hanging baskets ' either side** ... |
| | Mask Out | ... But it is able to carry just as much power - 400,000 volts . **It is designed to be less obtrusive and will be used for clean energy .** |
| *Omission* | Source | ... Opening statements in his trial are scheduled to begin Monday ... |
| | Reference | ... **Opening statements are scheduled Monday in the trial of James Holmes** ... |
| | Vanilla | ... Prosecutors are not swayed, will seek the death **penalty . Opening statements in his trial are scheduled to begin Monday . Holmes** says he was suffering ' a psychotic episode ' at the time ... |
| | Mask Emb | ... Prosecutors are not swayed, will seek the death **penalty . Opening statements in his trial are scheduled to begin Monday . Holmes** says he was suffering ' a psychotic episode ' at the time ... |
| | Mask Out | ... Prosecutors are not swayed and will seek the death **penalty . Holmes** says he was suffering ' a psychotic episode ' at the time ... |

Table 7: Examples from the CONLL correction task.

| | | |
|---|---|---|
| *Hallucination* | Source | They can become anyone . |
| | Reference | They can become **anyone** . |
| | Vanilla | They can become **anyone** . |
| | Mask Emb | They can become **friends with anyone** . |
| | Mask Out | They can become **anyone** . |
| *Omission* | Source | In conclude , people should think carefully of what is the consequences of telling the relatives his or her generic disorder issue . |
| | Reference | In conclusion , people should think carefully about what is the consequences of telling the relatives his or her **generic disorder issue** . |
| | Vanilla | In conclusion , people should think carefully about what is the consequences of telling the relatives his or her **generic disorder issue** . |
| | Mask Emb | In conclusion , people should think carefully about what is the consequences of telling the relatives his or her **generic disorder issue** . |
| | Mask Out | In conclusion , people should think carefully about what is the consequences of telling the relatives his or her **generic issue** . |

