# OpenReview forum: "Understanding and Improving Encoder Layer Fusion in Sequence-to-Sequence Learning"
_ICLR.cc/2021/Conference — ICLR 2021 Poster_

### Official Review · AnonReviewer1 · 2020-10-26
**Thorough analysis of encoder fusion asking for more focus on impact**

**Rating:** 5
**Confidence:** 5

**Review:**

The authors perform a thorough analysis of encoder fusion for Transformers: which encoder layer should the N-th decoder layer attend to? It turns out that the final decoder layers often attend to the encoder embeddings, leading the authors to provide them to the last decoder layer which leads to small improvements of performance on machine translation, summarization and grammar correction tasks. These are nice results, but the gains are small and the models are tested in the very basic configurations. These tasks and techniques, as well as some numbers used to claim state-of-the-art, are from a few years ago (e.g., SOTA on en-de translation is higher currently than the authors claim and higher than their number). It would be interesting to see if the presented conclusions hold for larger models - esp. for a T5 Transformer on masked language modeling, as this would be a more commonly used model in 2020. Unluckily, it is quite possible that increased activation size may negate the benefits of the authors' technique. It may well though make it even more important -- it would be really good to know! Lacking these experiments, we cannot recommend acceptance at this point.

---

> ### Author Response · Authors · 2020-11-24
> **Response to AnonReviewer1**
>
> 1. > *These tasks and techniques, as well as some numbers used to claim state-of-the-art, are from a few years ago (e.g., SOTA on en-de translation is higher currently than the authors claim and higher than their number).*
>
>   "Existing"  denotes the existing results using the same preprocessed datasets or preprocessing scripts, instead of claiming the SOTA results. To strengthen our Seq2Seq baselines on top of the existing models, we employed several simple yet effective techniques: 1) For the En-De and En-Fr translation tasks, we follow Ott et al. (2018) to train the models with the huge mini-batches of nearly 460K tokens, significantly improving the model performances. To clear up the misunderstanding of the existing score of En-De, we replace it with the result of the big-batch training (Ott et al., 2018) in Table 2; 2) For the summarization task, we follow Paulus et al. (2018) to disallow repeating the same trigram; 3) For the correction task, we apply a very strong regularization technique SwitchOut (Wang et al., 2018) to learn a better model. As far as we know, this is the first time to apply this technique to a correction task; 4) In particular, our Ro-En and En-Fr results are very competitive even with such a pure model architecture.
>
>   As listed in Table 2, our reimplemented baselines achieve better or comparable results with the existing results, demonstrating the reliability of the conclusions in this paper based on the stronger baselines.
>
>   Ott, Myle, Sergey Edunov, David Grangier, and Michael Auli. Scaling Neural Machine Translation. WMT 2018.
>   Paulus, Romain, Caiming Xiong, and Richard Socher. A Deep Reinforced Model for Abstractive Summarization. ICLR 2018.
>   Wang, Xinyi, Hieu Pham, Zihang Dai, and Graham Neubig. SwitchOut: an Efficient Data Augmentation Algorithm for Neural Machine Translation. EMNLP 2018.
>
> 2. > *It would be interesting to see if the presented conclusions hold for larger models - esp. for a T5 Transformer on masked language modeling, as this would be a more commonly used model in 2020. Unluckily, it is quite possible that increased activation size may negate the benefits of the authors' technique. It may well though make it even more important -- it would be really good to know! Lacking these experiments, we cannot recommend acceptance at this point.*
>
>   We respectfully disagree with the comment “increased activation size may negate the benefits of the authors' technique”. Larger models are not always the better models -- a suitable model size related to data scale is a better choice for Seq2Seq tasks. In our preliminary experiments, we tested several model variants for each Seq2Seq task and used its best-performing setting as the baseline: 1) Base Transformer (6 enc/dec layers and 512 hidden size) for the Ro-En, summarization, and correction tasks; and 2) Big Transformer (6 enc/dec layers and 1024 hidden size) for the En-De and En-Fr translation tasks.
>
>   We reimplemented the Big T5 model (12 enc/dec layers and 1024 hidden size), which failed to outperform our Transformer-Big baseline (28.2 vs. 28.9) when learning from scratch on the En-De dataset. The reproduced result (i.e. 28.2) is consistent with the results reported in their paper (i.e. 28.1/41.0 on the En-De and En-Fr datasets), which underperform our baselines (28.9/43.4).
>
>   Our proposed SurfaceFusion method can consistently improve the model performance across model variants with different sizes, demonstrating the effectiveness and universality of our approach. With the above clues, we kindly ask the reviewer to reconsider the decision.

---

### Official Review · AnonReviewer2 · 2020-10-27
**Comments on the paper**

**Rating:** 5
**Confidence:** 4

**Review:**

The paper proposes a *fine-grained layer attention* (FGLA) to analyze Encoder Fusion method.

While the introduction of FGLA allows for the analysis of the contribution of individual encoder layers, it changed the model’s architecture. Thus, when a model with FGLA is trained, it’s optimized differently such that the use of each separated layer in the encoder is modeled explicitly through attention weight. As a result, the contribution of each layer found in Seq2seq with FGLA might not be the same as in the standard Seq2seq. Therefore, I think the conclusion might not be appropriate for the standard Seq2Seq model. In other words, the source representation bottleneck hypothesis might not hold true for standard Seq2seq.

Compared to the previous fusion method that uses a scalar per layer as attention weight (i.e., *Transparent Attention*) in the work of Bapna et. al., 2018, FGLA generalizes it by making a vector per layer. This generalization is straightforward and simple. The later analysis in the paper is carried out on FGLA. However, I find that the same analysis can be performed with Transparent Attention. The introduction of FGLA seems not well-motivated in this regard. In terms of interpretability, Transparent Attention is more interpretable since there is one scalar per layer, whereas FGLA has to report average weight per layer (Figure 1).

For experimental results, FGLA is not compared with other fusion methods. Thus it’s unclear how the proposed method positions itself in the previous work on Encoder Fusion.

A previous work of ([Nguyen and Chiang, 2018](https://www.aclweb.org/anthology/N18-1031/)) proposed a similar way of fusing source embeddings directly to the output of the decoder. Their work can be seen as Surface fusion proposed in this paper.

While the motivation for this kind of fusion might be obvious for machine translation, it’s less so for summarization and grammatical error correction. I think it’s worth spending some text/examples on why this is needed for those tasks.


**References**
Improving Lexical Choice in Neural Machine Translation. Nguyen and Chiang, NAACL 2018

---

> ### Author Response · Authors · 2020-11-24
> **Response to AnonReviewer2 (Part 1)**
>
> 1. > *While the introduction of FGLA allows for the analysis of the contribution of individual encoder layers, it changed the model’s architecture. Thus, when a model with FGLA is trained, it’s optimized differently such that the use of each separated layer in the encoder is modeled explicitly through attention weight. As a result, the contribution of each layer found in Seq2seq with FGLA might not be the same as in the standard Seq2seq. Therefore, I think the conclusion might not be appropriate for the standard Seq2Seq model. In other words, the source representation bottleneck hypothesis might not hold true for standard Seq2seq.*
>
>   Although it is a potential threat, we have to introduce the FGLA to evaluate the contribution of each encoder layer. We believe that the bottleneck hypothesis of source representation still holds true for standard seq2seq. The difference between FGLA and the vanilla Transformer is the connections between encoder and decoder layers. If the source representation does not exist, FGLA would fail to improve the model performance, which turned out to be false in Tables 1&2 in the revised submission. These results reconfirm our hypothesis. Along this direction, we propose a novel SurfaceFusion for a better information flow for Seq2seq, which further improves the model performance.
>
> 2. > *Compared to the previous fusion method that uses a scalar per layer as attention weight (i.e., Transparent Attention) in the work of Bapna et. al., 2018, FGLA generalizes it by making a vector per layer. This generalization is straightforward and simple. The later analysis in the paper is carried out on FGLA. However, I find that the same analysis can be performed with Transparent Attention. The introduction of FGLA seems not well-motivated in this regard. In terms of interpretability, Transparent Attention is more interpretable since there is one scalar per layer, whereas FGLA has to report average weight per layer (Figure 1).*
>
>   We choose FGLA instead of Transparent Attention for two reasons. First, FGLA achieves better performance in our preliminary experiments (Table 1 in the revised submission). This is consistent with previous studies (Tu et al. 2017; Choi et al. 2018), which show that a weight vector outperforms its scalar counterpart. Second, the vector weight of FGLA enables the singular value analysis (Figures 3&4) to evaluate the expressivity of attended dimensions in the encoder embedding layer, which shows that the proposed SurfaceFusion method improves model performance by extracting useful surface features as expected. The singular value analysis cannot be done by the scalar weight of Transparent Attention method.
>
>   Zhaopeng Tu, Yang Liu, Zhengdong Lu, Xiaohua Liu, and Hang Li. Context Gates for Neural Machine Translation. TACL 2017.
>   Heeyoul Choi, Kyunghyun Cho, and Yoshua Bengio. Fine-Grained Attention Mechanism for Neural Machine Translation. Neurocomputing, 2018.
>
>
> 3. > *For experimental results, FGLA is not compared with other fusion methods. Thus it’s unclear how the proposed method positions itself in the previous work on Encoder Fusion.*
>
>   We respectfully point out that this is a misunderstanding. We have compared FGLA with several existing fusion methods, as listed in the last paragraph of Page 3 in the initial submission. Given one additional page, we show the results in Table 1 of the revised submission to improve clarity. FGLA consistently outperforms other fusion methods.

---

> > ### Author Response · Authors · 2020-11-24
> > **Response to AnonReviewer2 (Part 2)**
> >
> > 4. > *A previous work of (Nguyen and Chiang, 2018) proposed a similar way of fusing source embeddings directly to the output of the decoder. Their work can be seen as Surface fusion proposed in this paper.*
> >
> >   Thank you for pointing out this closely related work. Although both methods exploit source surface features to improve target predictions, there are considerable differences: SurfaceFusion performs feature fusion at probability-level, while their method fuses features at representation level. In addition, they evaluate their approach on the shallow RNN-based model, which may fail to work on deep Transformer models. To validate our hypothesis, we re-implemented their method on top of Transformer. Experimental results show that their method fails to outperform the vanilla Transformer model (33.3 vs. 33.8 on the Ro-En dataset). We will include the above discussion in the revised paper.
> >
> > 5. > *While the motivation for this kind of fusion might be obvious for machine translation, it’s less so for summarization and grammatical error correction. I think it’s worth spending some text/examples on why this is needed for those tasks.*
> >
> >   The aim of this paper is to understand why and when EncoderFusion works for general Seq2Seq tasks. We choose summarization and grammatical error correction tasks since they are representative Seq2Seq tasks. The machine translation task has distant input/output domains (i.e. in different languages), while the summarization and grammatical error correction tasks have similar input/output domains (i.e. in the same language). We believe that using the three tasks together is able to offer a more comprehensive understanding of the EncoderFusion method. The text/examples around Table 4 of §A.2 show that EncoderFusion alleviates prediction omission and hallucination in the summarization and grammatical error correction tasks.

---

### Official Review · AnonReviewer3 · 2020-10-28
**An interesting paper investigates how encoder layer fusion works in a multi-layer encoder**

**Rating:** 7
**Confidence:** 3

**Review:**

#####################   Summary   ####################

This paper introduces the fine-grained layer attention to evaluate the contribution of individual encoder layers and investigate how encoder layer fusion works, where the decoder layers have access to information for various encoder layers as opposed to only the final encoder layer as in standard Transformer.

Based on the observations that the encoder embedding layer is important to the success of encoder layer fusion and the uppermost decoder layer pays more attention to the encoder embedding layer, this paper proposes SurfaceFusion, which only connects the encoder embedding layer to the softmax layer of decoders, leading to quite substantial gains in metrics such as BLEU.

#####################    Strengths   ####################
1. This paper is really clearly written. The paper is easy to follow and understand.
2. The proposed SurfaceFusion is well-motivated by a series of experiments on different Seq2Seq tasks.
3. The experimental results are very solid.

#####################   Concerns   ####################

I hope the author can select several following concerns as feedback.

1. In this work, the authors find that in the summarization model, almost all decoder layers focus more on the encoder embedding layer. In other words, compared with the translation model and correction model, the summarization model tends to use more surface features from the encoder embedding layer. Therefore, it is reasonable to speculate that the summarization model will receive more improvements from the proposed SurfaceFusion. However, the experimental results in Table 1 show that the translation and correction models achieve higher improvements than the summarization model.

2. Figure 1 shows that the uppermost decoder layer also pays attention to other encoder layers, why SurfaceFusion only uses the encoder embedding layer? The authors claim that “although the layer attention model partially alleviates this problem, it potentially introduces unnecessary intermediate encoder representations.” Can you show some visualizations/experimental results to support this hypothesis that the intermediate encoder representations are unnecessary for the uppermost decoder layer?

3. The experiments show that the encoder embedding layer is beneficial for all decoder layers, why the proposed SurfaceFusion does not consider connecting the encoder embedding layer to all decoder layers. A much wider empirical investigation would be appreciated.

4. Along the same line, some relevant works are omitted [1][2][3][4][5][6], especially for the work from [1], which also considers the connections between the lower encoder layers and uppermost decoder layer, and boosts similar or even higher results on the same benchmarks (EN-DE and EN-FR) as this work. I would suggest the authors take a discussion and a comparison to that work.

5. The margin of some gains in Table 1 is small. Statistical significant test and error range are highly appreciated.

Overall, this paper is really clearly written, the proposed SurfaceFusion is novel and well-motivated, and experimental results are very solid. I recommend the authors to enrich this work by deepening their analysis. A much wider empirical investigation would be appreciated.

[1] Layer-Wise Cross-View Decoding for Sequence-to-Sequence Learning.

[2] Neuron interaction based representation composition for neural machine translation. In AAAI 2020.

[3] Dynamic layer aggregation for neural machine translation with routing-by-agreement. In AAAI 2019.

[4] Learning deep transformer models for machine translation. In ACL 2019.

[5] Dense information flow for neural machine translation. In NAACL-HLT 2018.

[6] Multi-layer representation fusion for neural machine translation. In COLING 2018.

#####################    After Rebuttal   ####################

I thank the authors for responding to the comments and have read them carefully.
The authors have addressed all my concerns in the rebuttal and I vote for acceptance of this submission.

---

> ### Author Response · Authors · 2020-11-24
> **Response to AnonReviewer3**
>
> 1. > *In this work, the authors find that in the summarization model, almost all decoder layers focus more on the encoder embedding layer. In other words, compared with the translation model and correction model, the summarization model tends to use more surface features from the encoder embedding layer. Therefore, it is reasonable to speculate that the summarization model will receive more improvements from the proposed SurfaceFusion. However, the experimental results in Table 1 show that the translation and correction models achieve higher improvements than the summarization model.*
>
>   While we do not give a complete explanation of this phenomenon. We conjecture that it is caused by the following reasons: 1) The summarization task is different from the other two tasks, since it required more "copy" operations from the input document to accomplish the task. Accordingly, the vanilla Transformer model may have learned to extract more surface features from layer-by-layer abstraction and residual connection than the other two tasks; 2) The improvement of the SurfaceFusion can not be reflected by the ROUGE metric; and 3) we employ a strong baseline (better than existing works, e.g. 37.2 vs. 36.8), thus it is difficult to improve upon it.
>
> 2. > *Figure 1 shows that the uppermost decoder layer also pays attention to other encoder layers, why SurfaceFusion only uses the encoder embedding layer? The authors claim that “although the layer attention model partially alleviates this problem, it potentially introduces unnecessary intermediate encoder representations.” Can you show some visualizations/experimental results to support this hypothesis that the intermediate encoder representations are unnecessary for the uppermost decoder layer?*
> 3. > *The experiments show that the encoder embedding layer is beneficial for all decoder layers, why the proposed SurfaceFusion does not consider connecting the encoder embedding layer to all decoder layers. A much wider empirical investigation would be appreciated.*
>
>   We give this claim based on the results of our preliminary experiments. We have validated the following variants of FGLA: 1) A simplified version of FGLA (FGLA-I) that connects the decoder output layer to only the encoder embedding and output layers, is on par with the vanilla FGLA that connects to all the encoder layers. We conclude that the intermediate encoder layers are less important for the representation fusion; 2) Another variant FGLA-II that connects all decoder layers to the encoder embedding and output layers, achieves similar performance with FGLA-I. This indicates that it may not be necessary to connect the encoder embedding layer to all decoder layers.
>
>   We include the above discussion in the revised paper to strengthen the motivation.
>
> 3. > *Along the same line, some relevant works are omitted [1][2][3][4][5][6], especially for the work from [1], which also consider the connections between the lower encoder layers and uppermost decoder layer, and boosts similar or even higher results on the same benchmarks (EN-DE and EN-FR) as this work. I would suggest the authors take a discussion and a comparison to that work.*
>
>   We include your suggested relevant works [1-6] in the sections of Introduction and Related Work in the revised submission. Our work differs from previous studies in that: 1) while most of them focus on improving model performance with SurfaceFusion, we aim to give a better understanding of SurfaceFusion and consolidate confiscated findings reported by previous studies; and 2) the design of our model is motivated by the findings of FGLA, while theirs are mainly based on research assumptions. In addition to our proposed fusion strategies, we believe that the understanding and findings revealed by our paper can motivate good follow-on works in a line of research on encoder layer fusion.
>
> 5. >*The margin of some gains in Table 1 is small. Statistical significance test and error range are highly appreciated.*
>
>   For translation tasks, our model significantly outperforms the baseline with p<0.01 on Ro-En and p<0.05 on En-De and En-Fr according to the significance method in Koehn (2004). For the task of summarization, the error ranges of three independent runs are 37.19 (±0.02) for baseline and 37.89 (±0.03) for SurfaceFusion. For the task of grammatical error correction, the error ranges of three independent runs are 53.40 (±0.93) for baseline and 55.78 (±0.77) for SurfaceFusion. As seen, our approach consistently outperforms the baselines in all cases.
>
>   Philipp Koehn. Statistical significance tests for machine translation evaluation. EMNLP 2004.

---

### Official Review · AnonReviewer4 · 2020-10-28
**Interesting idea, but I have questions**

**Rating:** 7
**Confidence:** 4

**Review:**

This is an interesting idea where the authors propose "SurfaceFusion", where they use the source embeddings learned by the encoder to modulate the output of the decoder at the final layer. The authors claim this is because the embeddings contain valuable information that is lost during encoder processing because the encoder lacks the capacity to represent both semantic and surface features. The authors then show through a series of experiments that attending over the encoder embeddings is useful, and propose a way to integrate the information from the embeddings directly into the last layer of the decoder, showing that this improves experimental results.

Few comments:

1) there are some grammatical and spelling errors, e.g.: "analyses", "which is the EncoderFusion expected to solve"
2) while the premise and analysis are interesting, i am curious about the reasons behind the design choice of "SurfaceFusion". If the encoder source embeddings are very important, and layerwise/finegrained attention perform similarly well, why not simplify the approach and simply add/concat them to the output of the encoder (or somewhere else where it makes sense)? have you tried this instead of introducing additional hyperparams?
3) what are the costs in terms of wall-clock time when introducing an additional softmax operation for every token in the decoder?

---
update:

Thanks for the clarifications. I've read the response and other reviews and have updated my rating.

---

> ### Author Response · Authors · 2020-11-24
> **Response to AnonReviewer4**
>
> 1. > *While the premise and analysis are interesting, i am curious about the reasons behind the design choice of "SurfaceFusion". If the encoder source embeddings are very important, and layerwise/finegrained attention perform similarly well, why not simplify the approach and simply add/concat them to the output of the encoder (or somewhere else where it makes sense)? have you tried this instead of introducing additional hyperparams?*
>
>   In our preliminary experiments, we have tried simply adding or concatenating encoder embedding to the encoder output, which fail to outperform the baseline model (28.2 / 27.9 vs. 28.9). By observing the learning curves, we found that these techniques brought a faster convergence at the beginning of training, but fell into a sub-optimal local optimum soon.
> One possible reason is that the model is biased in learning easy-to-learn surface features from the embedding layer and thus quickly overfits. Conversely, the proposed “SurfaceFusion” performs fusion at probability-level to strike the right balance between the learning of both surface and abstract features, which lead to better local optimum and model performance.
>
> ---
>
> 2. > *What are the costs in terms of wall-clock time when introducing an additional softmax operation for every token in the decoder?*
>
>   The additional softmax brings a 10% speed slower than vanilla model (from 0.018s per token to 0.020s per token). This is included in the revised paper.

---

### Author Response · Authors · 2020-11-24
**General Response to All Reviewers**

We thank the reviewers for the insightful comments and constructive suggestions, which will serve to improve the paper considerably. We will attend to all comments to the best extent. In the revised paper, we have accordingly made the following changes:
- Add a new Table 1 to clarify the comparisons with previous works.
- Discuss the other variants of FGLA.
- Add inference latency of SurfaceFusion.
- Discuss the other relevant works.
- Update Table 2 with new existing results for En-De and En-Fr.
- Fix typos

---

### Decision · Program_Chairs · 2021-01-07
**Final Decision**

**Decision:**

Accept (Poster)

**Comment:**

This paper proposes fine-grained layer attention to evaluate the contribution of individual encoder layers. This departs from the standard transformer architecture where the decoder uses only the final encoder layer. This paper investigates how encoder layer fusion works, where the decoder layers have access to information for various encoder layers. The main finding of the paper is that the encoder embedding layer is particularly important. They propose SurfaceFusion, which only connects the encoder embedding layer to the softmax layer of decoders, leading to accuracy gains.

There was some disagreement among reviewers about this paper. Overall, I found this a simple but effective contribution with interesting findings that can help future research in seq2seq models. Some of the weaknesses (discussing other relevant works, discussing other variants of FGLA, adding new experimental results) have been addressed in the updated version of the paper. One of the reviewers suggested running additional experiments on GLUE-style tasks (with a masked language model) to be really sure if the technique is convincing, and particularly try it with larger models (T5 was suggested). While adding those experiments would be a plus, I disagree that this is crucial - this paper is focusing on seq2seq tasks and is already considering several tasks: summarization, MT, and grammar correction. The results found by this paper are interesting and can foster future research extending this beyond these 3 tasks. Even if larger models can make the improvements smaller, there are many inconveniences in just increasing scale (memory consumption, energy consumption, etc.) It is my opinion that the community should value research that tries to understand the weaknesses of smaller models, rather than relying on large scale models to solve all problems.